# Systems biology approach identifies functional modules and regulatory hubs related to secondary metabolites accumulation after transition from autotrophic to heterotrophic growth condition in microalgae

**Bahman Panahi** [1] *, **Mohammad Farhadian** [2], **Mohammad Amin Hejazi** [3]

**1** Department of Genomics, Branch for Northwest & West Region, Agricultural Biotechnology Research Institute of Iran (ABRII), Agricultural Research, Education and Extension Organization (AREEO), Tabriz, Iran, **2** Department of Animal Science, Faculty of Agriculture, University of Tabriz, Tabriz, Iran, **3** Department of Food Biotechnology, Branch for Northwest & West Region, Agricultural Biotechnology Research Institute of Iran (ABRII), Agricultural Research, Education and Extension Organization (AREEO), Tabriz, Iran

* b.panahi@abrii.ac.ir, panahibahman@ymail.com

**Data Availability Statement:** All data are available from the ENA database (https://www.ebi.ac.uk/ena) with PRJNA289168 and PRJNA484804 ID.

## Abstract

Heterotrophic growth mode is among the most promising strategies put forth to overcome the low biomass and secondary metabolites productivity challenge. To shedding light on the underlying molecular mechanisms, transcriptome meta-analysis was integrated with weighted gene co-expression network analysis (WGCNA), connectivity analysis, functional enrichment, and hubs identification. Meta-analysis and Functional enrichment analysis demonstrated that most of the biological processes are up-regulated at heterotrophic growth condition, which leads to change of genetic architectures and phenotypic outcomes. WGNCA analysis of meta-genes also resulted four significant functional modules across logarithmic (LG), transition (TR), and production peak (PR) phases. The expression pattern and connectivity characteristics of the brown module as a non-preserved module vary across LG, TR, and PR phases. Functional analysis identified Carotenoid biosynthesis, Fatty acid metabolism and Methane metabolism as enriched pathways in the non-preserved module. Our integrated approach was applied here, identified some hubs, such as a serine hydroxymethyltransferase (SHMT1), which is the best candidate for development of metabolites accumulating strains in microalgae. Current study provided a new insight into underlying metabolite accumulation mechanisms and opens new avenue for the future applied studies in the microalgae field.

## Introduction

Microalgae have become attractive sources for metabolites, such as lipids, enzymes, polymers, toxins, and pigments production. These components can be utilized as adjuvant drugs, dietary

**Funding:** The work was supported by the Agricultural Biotechnology Research Institute of Iran (ABRII). The funders had no role in study design, data collection and analysis, decision to publish, or preparation of the manuscript.

**Competing interests:** The authors have declared that no competing interests exist.

supplements, seafood baits, cosmetics, tertiary wastewater treatment, and "green energy" [1]. Productions of above-mentioned metabolites have been optimized using the cultivation of microalgae on diverse media [2,3]. Microalgae are largely photosynthetic and cultivated in illuminated environments. However, biomass productivity and titers in photoautotrophic cultivation range from 0.055 to 0.061 g/ L/day at the laboratory scale. It is much lower for the industrial scale, and cannot meet demands of the global market [4].

To overcome the poor biomass accumulation, open ponds that mimic natural environments of the microalgae are alternative option for the most commercial microalgae [5]. These cultivation systems present relatively lower construction and operating costs and large ones can be constructed on nonagricultural lands [6]. Nevertheless, these cultivation systems have several disadvantages such as expensiveness in the harvesting step, continuous need to clean water; unfeasibility for the secondary metabolite production; and susceptibility to environmental conditions [6]. To control environmental parameters during cultivation, closed photobioreactors have been designed [7]. However, inefficient light dispersion and fouls biofilm development in long-term cultures, and high initial investment make them uneconomical for low-cost end-products [8].

Heterotrophic cultures in closed bioreactors are among the most promising strategies put forth to overcome this challenge. Heterotrophy is defined as the use of organic carbon such as acetate and glucose instead of $CO_2$ as substrate and energy sources [9]. The cost-effectiveness, relative simplicity of operations and daily maintenance are main advantages of the heterotrophic cultures. It has been shown that the heterotrophic cultivation improves the production yield significantly and provides a feasible approach for metabolites production at industrial level [10].

Differential accumulation patterns of lipids and carotenoids after the transition from photoautotrophic to heterotrophic cultures have been documented in the large body of research [4]. However, little is known about underlying metabolite accumulations mechanisms at different culture conditions in microalgae.

With the development of high-throughput transcriptome sequencing technologies, the genes involved in the metabolite biosynthesis in microalgae have been identified [11].

However, previous studies in this field have been typically focused on identifying differentially expressed genes and did not consider the degree of interconnection between genes, where genes with similar expression patterns may be functionally important [12]. Importance of the network-based approaches to elucidate transcriptional circuits of metabolic processes [13] and discovery of key regulators under different environmental conditions [14] have been highlighted. Co-expression analysis is based on the 'guilt-by-association' paradigm, which stipulates two genes displaying correlated expression patterns across different conditions [15]. To the best of our knowledge, co-expression analysis has not been yet applied in microalga.

In this study, we integrated meta-analysis with weighted gene co-expression network to identify functionally enriched pathways and network-centric genes associated with secondary metabolites production after transition to the heterotrophic growth condition in Chlorella microalgae.

## Materials and methods

### Data collection

European Nucleotide Archive (ENA) database was used as a source of RNA-seq data collection. Datasets with biological samples for both autotrophic- and heterotrophic growth conditions were collected. The datasets belonged to *Auxenochlorella prototothecoides*. The first dataset (PRJNA289168) contains 3 samples of *A. protothecoides* grown in photoautotrophic condition

(in Bristol's salts plus 0.1% (w/v) proteose peptone) and 9 samples of heterotrophic cultures (Bristol's salts plus 12.5% (v/v) Rainbow papaya juice). The photoperiod was 24 h: 0 h (light: dark).The second data set, PRJNA484804, contains 3 samples of photoautotrophic cultures. For photoautotrophic cultivation, cells were maintained in a photoautotrophic medium containing basal medium (0.7 g $KH_2PO_4$, 0.3 g K2HPO$_4$, 0.3 g $MgSO_4$ 7$H_2O$, 0.3 mg FeS-O$_4$_7$H_2O$, 0.01 mg vitamin B1, and 1 mL A5 trace mineral solution) adding 5 g/L glycine under illumination of 60 μmol photons·m$^{-2}$ s$^{-1}$. Moreover, this dataset contains 9 samples of heterotrophic cultures. For heterotrophic cultivation, cells were maintained in a heterotrophic medium containing basal medium adding 30 g/L glucose and 0.5 g/L glycine under dark condition [4].

In both data sets, samples were harvested with three biological replicate during the logarithmic growth phase in autotrophic growth mode. In heterotrophic growth condition, samples were harvested at logarithmic phase, transitional point between growth and metabolite biosynthesis phase hereafter we called transitional phase, and during peak of metabolite biosynthesis, production peak.

## Data preprocessing

Quality control of the raw reads was done using FastQC v0.11.5 (http://www.bioinformatics. babraham.ac.uk/projects/fastqc/). Then, low quality (below 30) reads were removed with Trimmomatic software v0.32 [16] using the following parameters: TRAILING:3, SLIDING-WINDOW:4:20, MINLEN: 40. Finally, reads were aligned to the *A. protothecoides* version ASM73321v1 reference genome (accessed from https://www.ncbi.nlm.nih.gov/genome) with TopHat2 version 2.0.12 [17] using the default parameter.

## Differential gene expression analysis and meta-analysis

Expression counts of the *A. protothecoides* gene annotation [18] were quantified for each sample using Bioconductor RSubread package version 1.6.5 [19]. Differential gene expression was analyzed using the Bioconductor DESeq2 package version 1.10.1 [20]. Comparisons were performed using Wald's test to determine the log2-fold change. Then, False discovery rate (FDR) [21] correction was used to account for multiple testing (p.adjust value cutoff of<0.05). Three comparisons (Auto vs. Logarithmic phase, Auto vs. Transition phase, and Auto vs. production phase at heterotrophic growth condition) were performed to identify DEGs.

To reduce the batch effects between the two datasets, the empirical Bayes algorithm was applied [22]. Then, common genes between two datasets were selected based on the transcriptome meta-analysis according to the prescribed in previous studies [11]. Finally, identified meta-genes subjected to additional analysis (Fig 1).

## Weighted Gene Co-Expression Analysis (WGCNA)

Scale-free weighted co-expression networks of meta-genes were constructed using the WGCNA algorithm implemented in R WGCNA package [23]. Briefly, similarity co-expression matrix was calculated with Pearson's correlation cor (i,j) for meta-genes. Then, adjacencies between meta-genes were calculated using soft threshold power beta function [24] using the following formula:

$$a_{ij} = \left( 0.5 * \left( 1 + \mathrm{cor(i,j)} \right) \right)^{\beta}$$

where $a_{ij}$ represents the adjacencies between DEGs as a connection strengths index.

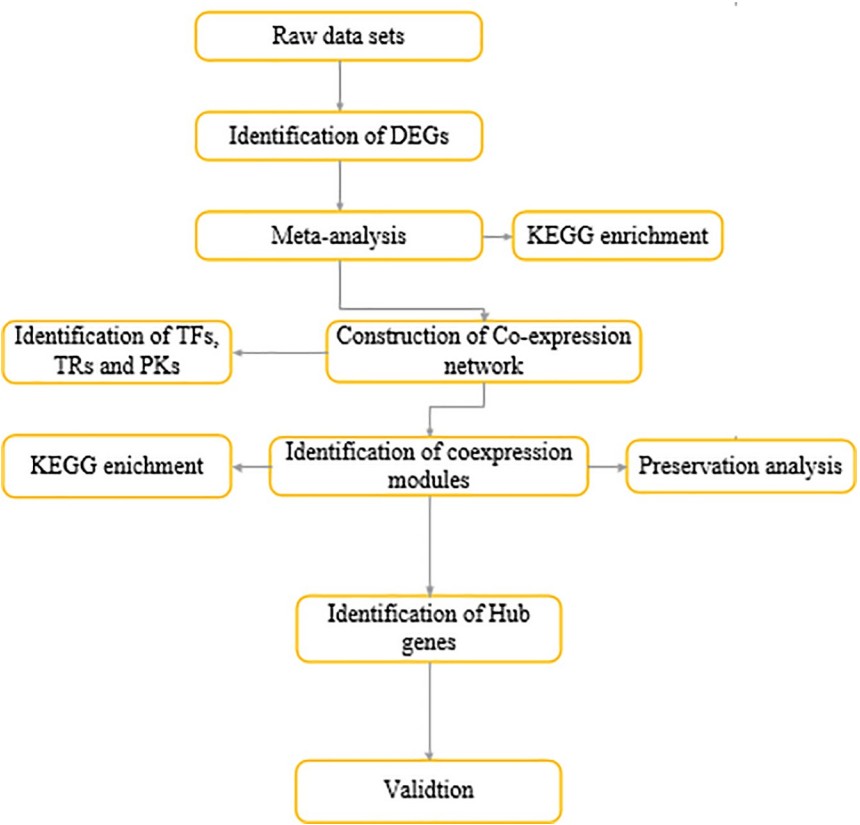

**Fig 1. Flow chart of applied systems biology approach in current study.**

In the power of beta = 12, linear regression model fitting index $R^2$ was higher than 0.8. Therefore, we selected this parameter as a satisfying scale-free topology criterion [12]. Finally, the adjacency was transformed into a topological overlap matrix (TOM) and corresponding dissimilarity matrix ($1-TOM$). Module identification was carried out with the dynamic tree cut method by hierarchically clustering genes using the dissimilarity matrix as the distance measure with a deep split value=2 and a minimum size cutoff= 60. The dissimilarity of eigen-genes (MEs) higher than 0.75, indicating the similar expression profiles of containing genes, were selected as a cut-off to merge initial modules. Module visualization and further analysis were performed with Cytoscape software.

To evaluate the module preservation after the transition to heterotrophic condition, module preservation function implemented in the WGCNA R package was applied. As prescribed in previous studies, Zsummary < 5 or medianRank > 8 was used as criteria for consideration of a module as a non-preserved module[25].

## Functional enrichment analysis

Pathway enrichment of initial sets of meta genes and non-preserved modules was performed using Kyoto encyclopedia of genes and genome (KEGG) database [26]. P-value < 0.05 was set as the cut-off criterion. Moreover, to identify potential transcription factors (TFs), transcriptional regulator (TRs), and protein kinases (PKs), amino acid sequences of meta genes were used to BLASTX search against the iTAK database version 1.6 [27] with a cut-off of $E \le 10^{-5}$.

### Identification and validation of hub genes

Hub genes were defined based on intra-module connectivity in non-preserved modules. Then, to validate and evaluate hub genes efficiency on different stages of discrimination, leave-one-out cross validation (LOOCV) was implemented based on expression values of hub genes [28].

## Results and discussion

### Transcriptomic changes after the transition to heterotrophic at different phases

Optimization of cultivation systems is one of the promising approaches to overcome the large-scale metabolite production challenges in microalgae. Dual-culture systems comprising heterotrophic growth followed by photoautotrophic cultivation are proposed as an effective issue for secondary metabolites production in some microalgae [29]. It has been proposed that multiple genes are involved in metabolites accumulation after the transition to heterotrophic growth modes, which may be interacted with each other to activate some signaling cascades [18]. Therefore, understanding the secondary metabolites accumulation underlying molecular mechanisms is important issues for optimization of this growth system to industrial levels production of secondary metabolites. Most of the previously studies only focus on screening of differentially expressed genes [4]. Nevertheless, it has been established that the integration of transcriptome meta-analysis with weighted co-expression network analysis is effective approach to shed light on complexity of biological processes. Using this approach, a set of responsible pathways and hub genes for secondary metabolites accumulation was determined that enabled us to propose some candidate genes for development of engineered microalgae strains. In this regards, 24 samples of RNA sequence data of *A. protothecoides* were retrieved from ENA database. The Meta genes abundance of *A. protothecoides* at three developmental stages (LG: logarithmic, TR: transition, and PR: production) during the transition from auto-trophic to heterotrophic cultures was analyzed using the deep transcriptome RNA-sequencing. Venn diagram was used to show the meta analysis results at three developmental stages after the transition to heterotrophic condition (Fig 2). Transcriptomic Changes at three developmental stages viz, LG, TR and PR were analyzed using the integrated systems biology approach (Fig 1).

As shown in Fig 2, at LG phase, 2962 meta genes between two datasets was identified which 1380 genes of them were down-regulated and 1582 of them were up-regulated after transition from auto- to heterotrophic condition (Fig 2A). It is whilst; the number of meta genes during the transition from auto to heterotrophic condition at theTR phase was increased to 3472 with 1666 down-regulated and 1806 up-regulated meta gene (Fig 2B). With the progress to the PR phase, meta DEGs between two data sets were 3577 which 1713 and 1864 of them is down and

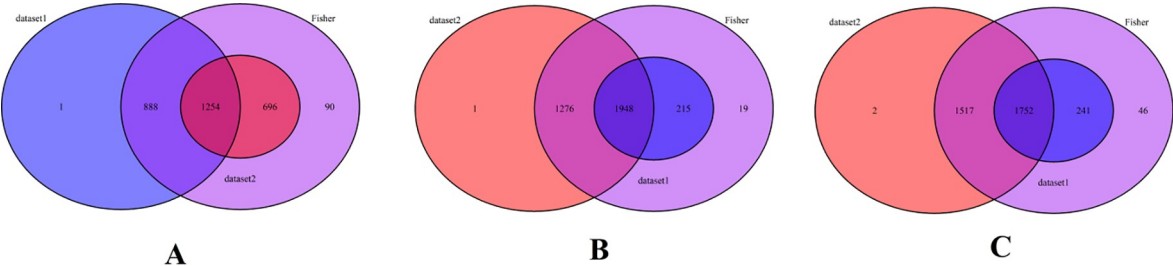

**Fig 2.** Identified meta genes in two data sets at logarithmic (A), transition (B) and production peak (C) phases after the transition from autotrophic to heterotrophic growth mode.

up regulated, respectively (Fig 2C, S1 Table). This result demonstrates that with the progress to developmental phases after the transition to heterotrophic growth condition, most of the biological processes is up-regulated, which lead to the change of genetic architectures and phenotypic outcomes such as secondary metabolites accumulation [4].

## Pathway enrichment of identified meta genes

The functional analysis of meta genes shows that some common and specific pathways are enriched at different phases (S2 Table). Results indicate that metabolic pathways are the main enriched pathways at developmental phases (S3 Table). In particular, a large number of genes were assigned to the metabolic pathways and biosynthesis of secondary metabolites. Although these pathways are found in all three phases, however, the changes in the number of up-regulated and down-regulated genes observed, suggesting the significant rearrangement in microalgae metabolism at different phases. Carbon metabolism is another enriched pathway. This observation confirms that of Martinez et al., [30], who showed that the metabolic pathways of carbon assimilation, size of the cells, volume densities of storage materials are changed after the transition to heterotrophic condition. Fatty acid biosynthesis pathway was found as another enriched pathway. In line with our finding, it has been reported that the increased amount of sugars in heterotrophic growth culture lead to accumulation of fatty acid [9].

Modulation of cellular energy state is another adaptation mechanism at heterotrophic cultures. Oxidative phosphorylation lead to produce ATP and NADH for maintenance and biosynthesis under the dark condition and carbon skeleton for biosynthesis under any growth condition [31]. Comparison of up-regulated genes involved in oxidative phosphorylation indicates that meta genes which encode the core subunits of mitochondrial complex I putatively, up-regulated at all phases. Nevertheless, some others which encode conserved non-core subunits of mitochondrial complex I, up-regulated specifically at PR phase. An example of non-core subunits which specifically up-regulated at the PR phase is ACP1 (encodes acyl carrier protein) a matrix-resident protein that has a key role in conformational regulation of mitochondrial complex I under different physiological condition [32]. It has been demonstrated that ACP homologs are key components in the production of polyketides and nonribosomal peptides [33], highlighting the roles of ACP in secondary metabolites accumulation at PR phases.

As expected, most of the photosynthesis-related meta genes such as PSAD-2 encoding the Photosystem I reaction center subunit II, PSBO1 encoding the Oxygen-evolving enhancer protein 1 of photosystem II, and PSAN encoding the Photosystem I reaction center subunit N are down-regulated at the LG, TR and PR phases. Down regulation of other photosynthesis-related genes at heterotrophic growth conditions has been previously demonstrated [4].

## Identification of the transcription factors (TFs), transcriptional regulators (TRs) and protein kinases (PKs)

TFs are master transcriptional regulators in different physiological conditions [34,35]. As shown in Fig 3 and S3 Table, SBP, C3H and MYB_related, GARP-G2-linke, C2C2-GATA, EP2/ERF-AP2, and bZIP are the top large classes of TFs, which are contributed in transcriptional regulation transcriptome circuits at LG, TR, and PR phases. Most of these transcription factors families, especially SBP and C3H families were reported to be up-regulated at secondary metabolites accumulation circumstances [36]. Therefore, engineering these TFs would be a potential approach to develop new strain with improved secondary metabolites accumulation ability. There is also evidence that SBP is implicated in redox clean up [37]. Moreover, many TFs related to photosynthetic carbon fixation such as MYB-related TFs is up-regulated with

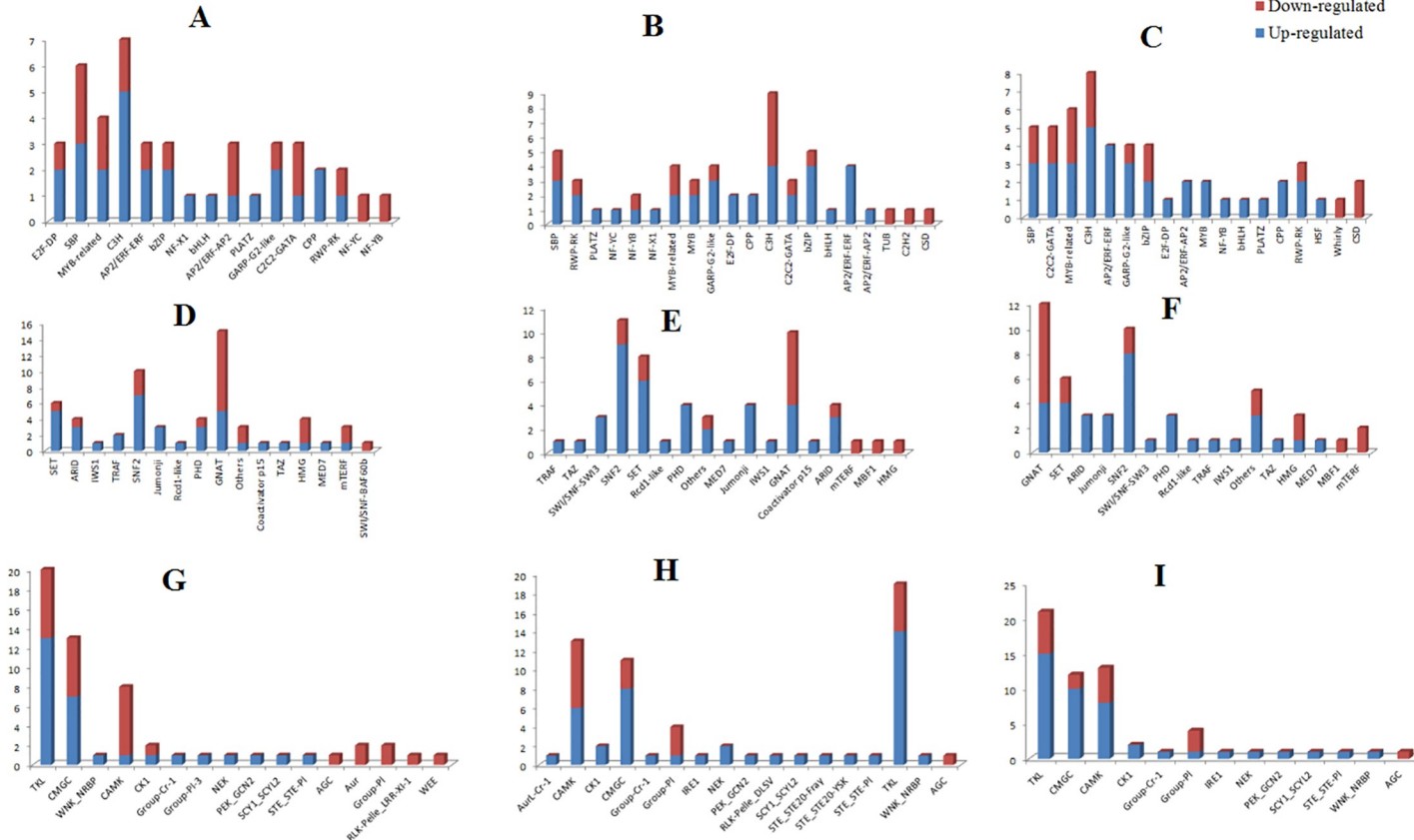

**Fig 3. Distribution of TFs, TRs, and PKs families identified in meta genes.** The number TF families (A, B and C), TRs families (D, E, and F), PKs families (G, H, and I) in meta genes at LG, TR and PR phases, respectively.

the same pattern across developmental phases. It has been demonstrated that Myb-related transcription factor is involved in phytochrome signal transduction pathway leading to up-regulation of lchb (light-harvesting chlorophyll b) gene in Arabidopsis [38,39]

Transcriptional regulators (TRs) are at the interface between sensing and responding to environmental conditions [40]. Results indicated that GNAT (GCN5-related-N-acetyltransferase), SNF2 (sucrose non-fermenting 2), and SET families are top classes of TRs at three phases (S3 Table). Interestingly, some TRs such as PHD shows different expression pattern at LG phase in comparison with TR and PR phases. It is apparent from data in Fig 3 that all of the TRs belong to PHD family are up-regulated at PR and TR phases. Prior study has indicated that SBP, bHLH, GNAT are involved in regulation of nitrogen assimilation, transportation and incorporation of ammonium into carbon skeletons via the glutamine synthetase/glutamate synthase cycle [41]. Moreover, a regulatory impact of SET families on major lipid droplet protein (MLDP1), which is contributed to TAG regulation, has been approved previously [41].

PKs play important roles in signaling networks including the perception of biotic agents, light quality and quantity, phytohormones, and various environmental conditions [42]. Fifty-seven (29 up-regulated and 28 down–regulated), sixty-two (43 up-regulated and 19 down-regulated), and fifty (43 up-regulated and 17 down-regulated) PKs were identified among the meta genes at LG, TR, and PR phases, respectively (S3 Table). Result illustrated that PKs mostly are up-regulated with the progress of PR phases at the heterotrophic conditions. Moreover, TKL (Tousled protein kinase), CMGC, and CAMK families are among the main

contributed PKs at heterotrophic conditions (Fig 3). The genes encoding TKL was dominantly up-regulated under the heterotrophic condition. As evidenced by genetic analyses of *Arabidopsis* mutant, TKL group play important role in the regulation of ethylene signaling [43] and accumulation of secondary metabolites [44] at heterotrophic condition.

## Identification of functional modules related to developmental phases

The details of the signed weighted gene co-expression network construction are prescribed in [45]. By using the steps described in Materials and Methods section, three separate networks of highly correlated meta genes at LG, TR and PR phases were constructed. Shortly, weighted adjacency matrices were created using the soft threshold power. In this study soft threshold power =9 was selected. Then, transformed dissimilarity matrices was applied for average linkage hierarchical clustering with method implemented in flashClust R package [46]. The dendrograms obtained from the preliminary analysis are shown in Fig 4. The structures of the dendrograms are changed at different phases after transition to the heterotrophic growth mode. To further inspection of the network pattern changes at different developmental phases, modules of dendrogram were defined using the dynamic branch cutting method, which is displayed by different colors (Fig 5). Using the mentioned parameter setting, the branch-cutting method resulted 15 co-expressed modules with an average size 235, 12 of which were significant. After generating co-expressed modules, we compared the results with the corresponding set of modules in different developmental phases. As shown, four modules were identified in the LG network. The yellow and turquoise modules as the smallest and largest modules contained 212 and 810 genes, respectively. In the TR co-expression network, seven modules were assigned (Fig 5) with a size range from 72 (black) to 1,422 (turquoise) genes. At the PR phase, four modules were identified with a size range from 223 genes (yellow modules) to 1053 genes (turquoise module). Preservation analysis showed that three modules viz, blue, turquoise, and yellow has persevered between three phases. However, the brown module showed non-preservation between different phases (Table 1). Since the connectivity and expression patterns of

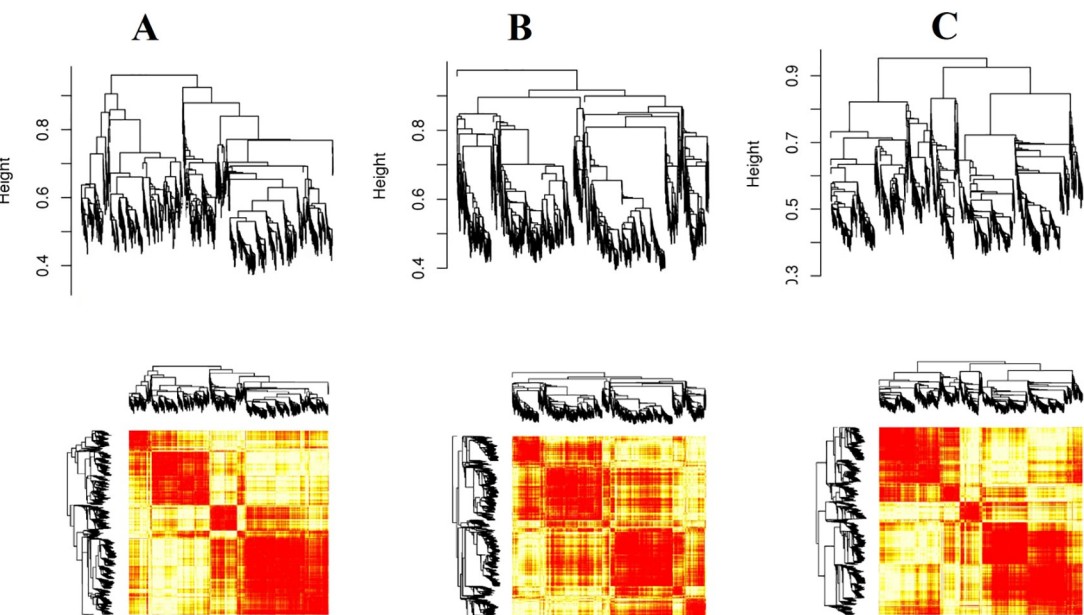

**Fig 4.** Hierarchical cluster constructed with WGCNA at LG (A), TR (B), and PR (C) phases. Each vertical line (leaf) represents the corresponding genes.

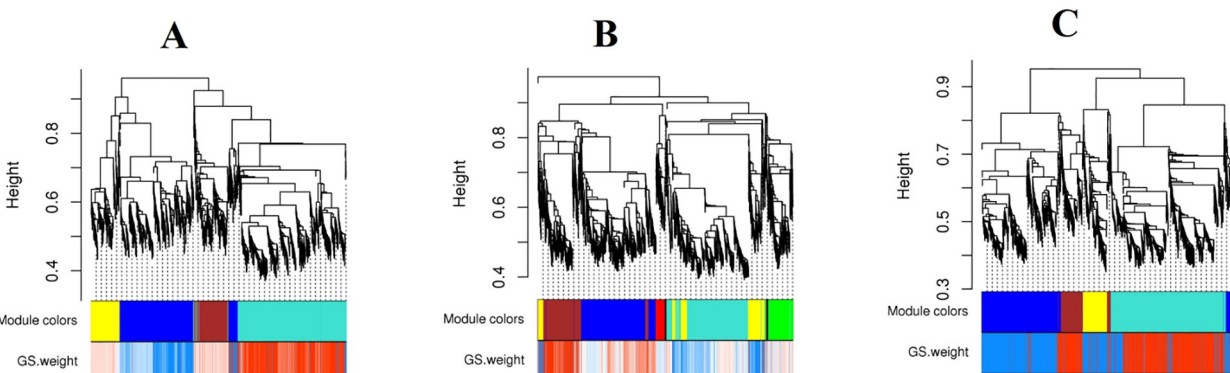

**Fig 5.** Visual representation of the changes in the module structure between LG (A), TR (B), and PR (C) phases. Modules are illustrated with different colors.

the co-expressed genes at non-preserved modules are altered [25], they may be related to the over-production of secondary metabolites at the PR phase. Because Grey and Gold modules contain the genes that not assigned to any modules, these modules were excluded from the analysis. Corresponding genes for each defined modules were presented in S4 Table.

## Functional annotation of non-preserved module

To unravel the potential mechanisms responsible for the accumulation of secondary metabolites at heterotrophic growth condition we focused on the non-preserved module. As presented, the brown module was determined as a non-preserved module in three phases. In the non-preserved module, expression patterns and connectivity characteristics of the genes vary across three phases (Fig 6). Functionally enrichment analysis identified "Biosynthesis of secondary metabolites" as a functionally enriched KEGG pathway in the brown module. Other significantly enriched pathways in brown module were "Carotenoid biosynthesis", "Fatty acid metabolism", "Methane metabolism", "Ascorbate and aldarate metabolism", "Peroxisome proliferator-activated receptors (PPARs) signaling pathway" and "Adipocytokine signaling pathway" (Table 2). Prior studies have also highlighted the variation of the carotenoid biosynthesis, fatty acid metabolism and Pentose phosphate pathway underlying genes in the autotrophic-heterotrophic cultivation transition culture mode [4,47,48]. However, hub genes identification and their connectivity have not been surveyed. There are several genes shared by several pathways such as ACS2 (Acetyl CoA synthetase) (eight pathways), LACS7 (Long-chain acyl-CoA synthetase 7) (four pathways), SHMT1 (Serine hydroxymethyltransferase) (four pathways). This also validates the relevance of the selected module for the detection of important players in metabolite accumulation [12]. Interestingly, Protein- Protein interaction network (PPI) analysis and connectivity measurement using a soft connectivity algorithm confirms that these genes are highly connected in the PR phase generated from the brown module, while there is

**Table 1. Details of conservation analysis of defined modules at different developmental phases with permutation=200.**

| Module name | Module size | medianRank | Zsummary |
|---|---|---|---|
| Blue | 469 | 2 | 32.56 |
| Brown | 184 | 8 | 4.26 |
| Turquoise | 681 | 3 | 34.39 |
| Yellow | 155 | 1 | 19.05 |

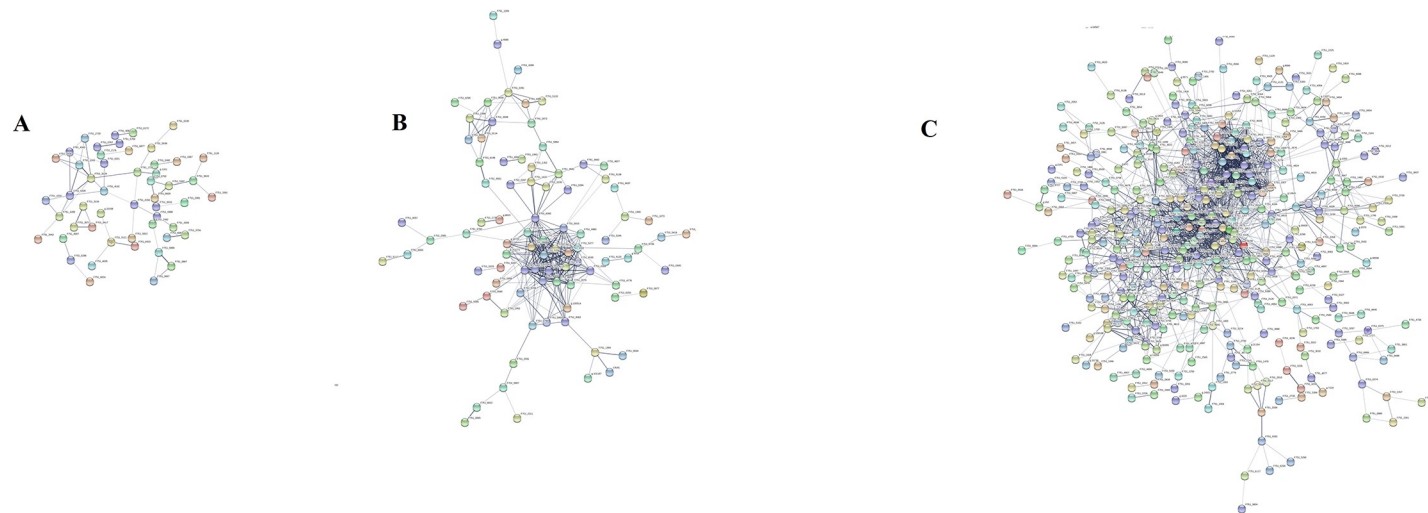

**Fig 6.** PPI networks of co-expressed meta genes in brown module at LG (A), TR (B), and PR (C) phases. Changes in intra-module connectivity are highlighted at different phases.

**Table 2. Functional enrichment of non-preserved module based on KEGG database.**

| KEGG Pathway | p-value | Genes |
|---|---|---|
| **Biosynthesis of secondary metabolites** | 0.0003 | ATHMT1,HMT1,ATSK1,SK1,FBA2,HEMA1, AtGNA1,GNA1,GME,CSY2,HEMC, CHLM,CLA,CLA1,DEF, DXPS2,DXS, BCE2,DIN3,LTA1,LYC,ALB1V,ALB1,CHLD,PDE166,V157,DHNS,ECHID,PORA, DHDPS2,HCEF1, ACS2,PDE226,PDS,PDS3,ATCAO,CAO,CH1,ALDH2B,ALDH2B7,SHM1,SHMT1,STM,AGT,AGT1,SGAT, EMB2778,FKP1,HMGS,MVA1,HISN5B,  CCR2,CRTISO,NOL,AtSS2,SS2,HEMG1,PPO1,PPOX,DELTA,OAT, ACSF,CHL27,CRD1,PFK5,ATUGD1,UGD1,LUT2,PCK1,PEPCK,CAC2,EMB2728,RPE |
| **Photosynthesis** | 4.4304E-7 | CF0 ATP synthase subunit II precursor,PSAO,OE23,OEE2,PSBP-1,PSII-P,PETE1,PSAF,PPL1,PSAL,PSB28,PSAD-2,PETC,PGR1,PSB27 |
| **Methane metabolism** | 0.0175 | FBA2,HCEF1,ACS2,SHM1,SHMT1,STM,AGT,AGT1,SGAT,HCEF1,PFK5 |
| **Carbon fixation** | 0.0009 | FBA2,RBCS1A,HCEF1,GAPB,SBPASE,PRK,HCEF1,SBPASE,PCK1,PEPK,EMB2728,RPE |
| **Microbial metabolism in diverse environments** | 0.0226 | FBA2,RBCS1A,CSY2,DHDPS2,HCEF1,ACS2,ALDH2B,ALDH2B7,SHM1,SHMT1,STM,GAPB,AGT,AGT1,SGAT, SBPASE,MLS,PRK,HCEF1,PFK5,SBPASE,PCK1,PEPCK,CAC2,EMB2728,RPE,MLS |
| **Porphyrin and chlorophyll metabolism** | 0.0003 | HEMA1,HEMC,CHLM,ALB1V,ALB1,CHLD,PDE166,V157,PORA,ATCAO,CAO,CH1,NOL,HEMG1,PPO1, PPOX,ACSF,CHL27,CRD1 |
| **Glyoxylate and dicarboxylate metabolism** | 0.0075 | RBCS1A,CSY2, SHMT1, AGT,SGAT,MLS |
| **Ascorbate and aldarate metabolism** | 0.0058 | GME,MIOX1, ALDH2B, MDAR1, UGD1 |
| **Valine, leucine and isoleucine degradation** | 6.7763E-10 | BCE2,DIN3,LTA1,MCCB, ALDH2B, ALDH2B7,MCCA, FKP1, CHY1, MCCB |
| **Peroxisom** | 0.0063 | IBR1,SDRA,ATLACS7,LACS7,ATLACS7,LACS7,ATDCI1,DCI1,AGT,AGT1,SGAT,DECR,SDRB,ACX4,ATG6, ATSCX |
| **Fatty acid metabolism** | 0.0211 | ACS2,LACS7, ALDH2B,ALDH2B7, ACX4,ATG6,ATSCX |
| **PPAR signaling pathway** | 0.0103 | ATLACS7,LACS7, GLI1,NHO1,ACX4,ATG6,ATSCX |
| **Adipocytokine signaling pathway** | 0.0042 | TOR,AS2,LACS7, F751_2317 |
| **Protein export** | 0.0065 | PLSP1, ALB3,CPFTSY,FRD4,APG2,PGA2,TATC,UNE3,54CP,CPSRP54,FFC,SRP54CP, AGY1,AtcpSecA |
| **Bacterial secretion system** | 7.5531E-6 | ALB3,CPFTSY,FRD4,APG2,PGA2,TATC,UNE3,54CP,CPSRP54,FFC,SRP54CP,AGY1,AtcpSecA |
| **Carotenoid biosynthesis** | 0.0097 | LYC,PDE226,PDS,PDS3,CCR2,CRTISO,LUT2 |
| **Pantothenate and CoA biosynthesis** | 0.0472 | ATHAL3,ATHAL3A,HAL3,HAL3A,PYD2 |
| **Mismatch and base excision repair** | 0.0117 | ATMLH1,MLH1,ATLIG1,LIG1,POLD3,ATPCNA1,PCNA1,RFC3,RFC5,EMB2780 |
| **Synthesis and degradation of ketone bodies** | 0.0372 | EMB2778,FKP1,HMGS,MVA1, g.47156 |

low or no such connection were found in the corresponding module at LG and TR phases. The potential of nodes with high intra-module connectivity as key and determinant genes in the different biological processes has been approved previously [25]. Moreover, it has been suggested that important nodes in large networks are often not among the whole-network hubs and selection of the hubs in sub-networks (modules) is more efficient than a whole-network. Based on these findings, we hypothesized that the nodes with high intra-module connectivity may play important roles in secondary metabolites over-production at the PR phase after the transition to heterotrophic conditions.

ACS2 encoding a chloroplastic acetyl-CoA synthetase catalyzes the conversion of acetate to acetyl-CoA. Experimental evidence has proved that ACS is a key enzyme in the biosynthesis of acyl glycerides [49]. Moreover, ACS allows consumption of the acetate and increase the carbon flux towards the synthesis of fatty acids to enhance biosynthesis and accumulation of lipid at fermentative growth condition [50]. High connectivity of this gene in the PR network and increased expression levels in the PR phase, may be highlighted the importance of ACS in lipids accumulation at heterotrophic conditions.

Another interesting hub gene that is part of the tightly connected cluster in the brown module is SHMT1, which encodes the Serine hydroxymethyltransferase. SHMT1 simultaneously catalyzes the reversible conversions of l-serine to glycine and tetrahydrofolate to 5,10-methylene tetrahydrofolate. Contribution of tetrahydrofolate in cellular one-carbon (C1) pathways and ROS generation at stress condition has been reported previously [51]. Moreover, prior study have proved that SHMT1 switch nitrogen and carbon metabolism to secondary metabolite biosynthesis [11], validating the results of current study. An earlier study also showed that the overexpression of the SHMT1 increases biomass production [52].

There are other interesting hub genes (top 30 hub genes) in the brown module, which are presented in S5 Table. We found that these genes are related to ribosome biogenesis, fatty acid metabolism, carotenoid biosynthesis, sulfur relay system.

Ribosome biogenesis is a central process in the growing cells. A recent study has indicated the translational regulation of ribosome biosynthesis in response to carbon depletion [53]. Among different ribosome biogenesis involved genes, EIF3I was demined as a key gene for metabolite accumulation, based on the connectivity analysis. In agreement with our findings, the core functionality of eIF3 has been confirmed in a previous study [54]. At heterotrophic growth condition, the inorganic carbon is sufficiently supplied and marked increasing of ribosome biogenesis is needed to the biosynthesis of proteins and metabolites [30].

Further analysis of connectivity in the none-preserved module at the PR phase reveals another gene, LACS7, a part of peroxisome proliferator-activated receptors (PPAR) and Adipocytokine signaling pathways [55], as another important gene. Newly synthesized free fatty acids (FAs) need to activate to acyl-CoA form by LACS to involves in the glycerolipid metabolic pathways, indicating the critical roles of LACS in FA metabolism [56]. In Arabidopsis, a family of nine genes encoding LACS protein has been identified to play their roles in various aspects of lipid metabolism at different subcellular localizations. Among these LACS isoforms, peroxisomal LACS6 and LACS7 are involved in fatty acid β-oxidation [57]. The results of our analysis indicated that LACS7 is up-regulated during the LG and TR phases. However; marked down-regulation is shown at the PR phase. Consequently, inhibition of β-oxidation would prevent the loss of TAG during the PR phase. These findings clearly show that the lipids accumulations are dominantly related to decreased lipid catabolism at the PR phase. In agree with our findings, it has been reported that inactivation of the LACS6 and LACS7 lipid breakdown [58]

Another interesting gene that shows a large difference between connectivity in PR versus LG and TR brown module is LCE (Lycopene epsilon cyclase) which is involved in the carotenoids biosynthesis. Transcriptional variation of LCE during the lutein and beta-carotene

accumulation circumstances has been reported [11] [4]; however, the changes in the connectivity pattern have not shown up yet, highlighting the efficiency of our integrative approach.

Some identified hub genes such as F751_5997, F751_3508, and F751_4289 were not annotated, whereas they should be considered as potential candidates for future studies. As abovementioned, the efficiency of hub genes in different phase's discrimination was assessed using the LOOCV method. Identified hub genes have discriminated different phases with 90.48% accuracy, validating the identified hub genes (S1 Fig).

## Conclusion

In summary, our results show that the integration of transcriptome meta-analysis with WGCNA, along with connectivity analysis and functional enrichment can be used to identify modules associated with secondary metabolite accumulation in microalgae. Identified modules are used for exploratory analysis of contributed pathways in metabolite accumulation, as exemplified by the identified pathways such as "Methane metabolism", "Ascorbate and aldarate metabolism" and " PPARs". Moreover, the integrated approach was applied here, proposed some candidate target genes, such as SHMT1, for the development of metabolites accumulating strains in microalgae. However, future efforts should be made to further investigation of the modules and candidate genes associated with secondary metabolites accumulation.

## Supporting information

**S1 Fig. Validation of hub genes based on LOOCV method.**
(TIF)

**S1 Table. Identified meta genes in three phase of transition from auto- to heterotrophic condition.**
(XLSX)

**S2 Table. Significant enriched pathways in different phases after transition to heterotrophic growth condition.**
(XLSX)

**S3 Table. Identified TFs, PKs and TRs across LG, TR and PR phases.**
(XLSX)

**S4 Table. Genes of each corresponding modules across LG, TR and PR phases.**
(XLSX)

**S5 Table. Hub genes and connectivity in brown module at PR phase.**
(XLSX)

## Acknowledgments

We would like to thank Dr Asgar Panahi for reading of the manuscript.

## Author Contributions

**Conceptualization:** Bahman Panahi.

**Data curation:** Bahman Panahi, Mohammad Farhadian.

**Formal analysis:** Bahman Panahi.

**Writing – original draft:** Bahman Panahi.

**Writing – review & editing:** Bahman Panahi, Mohammad Farhadian, Mohammad Amin Hejazi.

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
