## [Decision Letter · Decision Letter 0]

14 Jan 2020

PONE-D-19-31134

Systems biology approach identifies functional modules and regulatory hubs related to secondary metabolites accumulation after the transition from autotrophic to heterotrophic growth condition in the microalga

PLOS ONE

Dear Dr. Panahi,

Thank you for submitting your manuscript to PLOS ONE. After careful consideration, we feel that it has merit but does not fully meet PLOS ONE’s publication criteria as it currently stands. Therefore, we invite you to submit a revised version of the manuscript that addresses the points raised during the review process.

Please consider particularly the recommendations by Reviewer 2 who has identified some clarifications needed about some of your analytical procedures and interpretations.

We would appreciate receiving your revised manuscript by Feb 28 2020 11:59PM. To enhance the reproducibility of your results, we recommend that if applicable you deposit your laboratory protocols in protocols.io, where a protocol can be assigned its own identifier (DOI) such that it can be cited independently in the future. For instructions see: http://journals.plos.org/plosone/s/submission-guidelines#loc-laboratory-protocols

We look forward to receiving your revised manuscript.

Kind regards,

O. Roger Anderson

Academic Editor

PLOS ONE

Additional Editor Comments (if provided):

Both reviewers recommend that your interesting study has merit for publication, but also have noted some revisions that are recommended to improve some of the report of analyses and interpretations.

Journal Requirements:

 [No].

Please provide an amended Funding Statement that declares *all* the funding or sources of support received during this specific study (whether external or internal to your organization) as detailed online in our guide for authors at http://journals.plos.org/plosone/s/submit-now.  

Please state what role the funders took in the study.  If any authors received a salary from any of your funders, please state which authors and which funder. If the funders had no role, please state: "The funders had no role in study design, data collection and analysis, decision to publish, or preparation of the manuscript."

6. Thank you for stating the following in your Competing Interests section: 

[No].

Reviewers' comments:

Reviewer's Responses to Questions

**Comments to the Author**

1. Is the manuscript technically sound, and do the data support the conclusions?

Reviewer #1: Yes

Reviewer #2: Partly

2. Has the statistical analysis been performed appropriately and rigorously? 

Reviewer #1: Yes

Reviewer #2: Yes

3. Have the authors made all data underlying the findings in their manuscript fully available?

Reviewer #1: Yes

Reviewer #2: Yes

4. Is the manuscript presented in an intelligible fashion and written in standard English?

Reviewer #1: Yes

Reviewer #2: Yes

5. Review Comments to the Author

Reviewer #1: The manuscript entitled “Systems biology approach identifies functional modules and regulatory hubs related to secondary metabolites accumulation after the transition from autotrophic to heterotrophic growth condition in microalga” by Panahi et al., attempted to integrate the weighted gene co-expression network analysis (WGCNA) with transcriptome meta-analysis, connectivity analysis, functional enrichment, and hub genes to uncover the key metabolic targets underpinning the key biochemical pathways in microalgae. Though there are previous studies available in various microalgae including heterotrophic Auxenochlorella protothecoides (PubMed ID: 25012212), Asterarcys sp. (Li et al., 2020, Algal Research, vol. 45, 101753), here the authors used an integrated approach to overcome the existing hurdles. Though the study seems to be interesting, the following comments may be addressed.

1. The methods section is vague. What was the screening condition as mentioned in the data collection? It was stated that authors retrieved dataset PRJNA484804 which contains "3 samples of photoautotrophic cultures" and "9 samples of heterotrophic cultures", which is unclear, specifically about the heterotrophic cultures. Thus, the section may be rewritten.

2, It is not clear whether the authors considered a study network with more different experimental groups to expand the number of independent samples.

3, The authors could compare the current systems biology approach to the previous RNA-seq transcriptomic profiling result to show the highly significant positive correlation over all, also its limitations and strengths.

Minor comments:

1. For photoautotrophic culture, the authors shall mention the dark:light regime for algal cultivation (L. No. 82).

2. Unit for illumination shall be mentioned as μmol photons m−2 s−1 instead of lux (L. No. 83).

3. It is well known that growth phase during the cultivation period varies among the microalgae, thus it is necessary to provide the transition period between log and metabolite accumulation phase (L. No. 87).

Reviewer #2: WGCNA has powerful pontetial to search for genes of interests and even Hub genes for organims adapt to some environmental traits. Very limited studies were done in cyanobacteria and microalgae. Thus this ms had a very intertesting try. With two gene expression datasets, the authors managed to target some hub genes, which difinitely attract interests in the fields of microalgae and environmental sciences to students, researchers and investigators. However, there are several technique concerns about this study. First of all, two sets of data were obtained through different experiments and thus the basis for comparison and integration of these two data would be a questionable issue. The reliablity of the results would be discounted becuase of their different experiment conditions. Another minor issue about the WGCNA results, we nornally could obtained some significant modules, in colors and co0efficient factors with p-values. If have any, please provide and analyze. If not, please also explian.

6. PLOS authors have the option to publish the peer review history of their article (what does this mean?). If published, this will include your full peer review and any attached files.

Reviewer #1: No

Reviewer #2: Yes: Jiangxin WANG

---

## [Author Response · Author response to Decision Letter 0]

21 Jan 2020

Dear Reviewers 

Thanks a lot for your suitable comments. We revised the manuscript, hope to be satisfied

---

## [Decision Letter · Decision Letter 1]

23 Jan 2020

Systems Biology Approach Identifies Functional Modules and Regulatory Hubs Related to Secondary Metabolites Accumulation after Transition from Autotrophic to Heterotrophic Growth Condition in Microalgae

PONE-D-19-31134R1

Dear Dr. Panahi,

We are pleased to inform you that your manuscript has been judged scientifically suitable for publication and will be formally accepted for publication once it complies with all outstanding technical requirements.

With kind regards,

O. Roger Anderson

Academic Editor

PLOS ONE

Additional Editor Comments (optional):

Thank you for promptly attending to your revisions. The reviewers recommend acceptance of your revised manuscript.

Reviewers' comments:

Reviewer's Responses to Questions

**Comments to the Author**

1. If the authors have adequately addressed your comments raised in a previous round of review and you feel that this manuscript is now acceptable for publication, you may indicate that here to bypass the “Comments to the Author” section, enter your conflict of interest statement in the “Confidential to Editor” section, and submit your "Accept" recommendation.

Reviewer #2: All comments have been addressed

2. Is the manuscript technically sound, and do the data support the conclusions?

Reviewer #2: Yes

3. Has the statistical analysis been performed appropriately and rigorously? 

Reviewer #2: Yes

4. Have the authors made all data underlying the findings in their manuscript fully available?

Reviewer #2: Yes

5. Is the manuscript presented in an intelligible fashion and written in standard English?

Reviewer #2: Yes

6. Review Comments to the Author

Reviewer #2: The ms has been improved in the revised version. The concerns in my previous review were solved and added in the main text. I think now it is good to go.

7. PLOS authors have the option to publish the peer review history of their article (what does this mean?). If published, this will include your full peer review and any attached files.

Reviewer #2: Yes: Jiangxin WANG

---

## [Editor Report · Acceptance letter]

27 Jan 2020

PONE-D-19-31134R1 

Systems Biology Approach Identifies Functional Modules and Regulatory Hubs Related to Secondary Metabolites Accumulation after Transition from Autotrophic to Heterotrophic Growth Condition in Microalgae 

Dear Dr. Panahi:

I am pleased to inform you that your manuscript has been deemed suitable for publication in PLOS ONE. Congratulations! Your manuscript is now with our production department. 

With kind regards,

on behalf of

Dr. O. Roger Anderson 

Academic Editor

PLOS ONE